# Gating mechanism of hyperpolarization-activated HCN pacemaker channels

Rosamary Ramentol[1], Marta E. Perez[1] & H. Peter Larsson [1✉]

Hyperpolarization-activated cyclic nucleotide-gated (HCN) channels are essential for rhythmic activity in the heart and brain, and mutations in HCN channels are linked to heart arrhythmia and epilepsy. HCN channels belong to the family of voltage-gated $K^+$ (Kv) channels. However, why HCN channels are activated by hyperpolarization whereas Kv channels are activated by depolarization is not clear. Here we reverse the voltage dependence of HCN channels by mutating only two residues located at the interface between the voltage sensor and the pore domain such that the channels now open upon depolarization instead of hyperpolarization. Our data indicate that what determines whether HCN channels open by hyperpolarizations or depolarizations are small differences in the energies of the closed and open states, due to different interactions between the voltage sensor and the pore in the different channels.

[1] Department of Physiology and Biophysics, University of Miami, Miami, FL 33136, USA. ✉email: PLarsson@med.miami.edu

Hyperpolarization-activated cyclic nucleotide-gated (HCN) channels are essential for rhythmic activity in the heart and brain, and mutations in HCN channels are linked to heart arrhythmia and epilepsy[1,2]. HCN channels are activated by hyperpolarization (i.e., when the membrane potential becomes more negative), whereas Kv channels are activated by depolarization (i.e., when the membrane potential becomes more positive)[3–6]. Hyperpolarization-activated HCN channels and depolarization-activated Kv10.1 (EAG) channels have very similar tetrameric structures with six transmembrane segments (S1–S6) per subunit: S1–S4 form the voltage-sensing domain (VSD) and S5–S6 form the pore domain (PD)[7,8]. In both Kv and HCN channels, S4 is the positively charged voltage sensor[3,9–11] and the C-terminal part of S6 forms the gate[7,12–14]. In domain-swapped Kv channels (e.g., in Kv1.2 channels in which the VSD interacts with the PD in the neighboring subunit), it is assumed that the covalent S4–S5 linker transduces voltage-induced S4 movement to gate opening by pulling on the lower S6, as if the channels open by an electro-mechanical mechanism[8,15]. However, cutting the S4–S5 linker in non-domain-swapped channels, such as hyperpolarization-activated HCN channels or depolarization-activated Kv10.1 (EAG) and Kv11.1 (hERG) channels[7,8], does not prevent gating[16,17], showing that non-domain-swapped channels do not require a covalent link between S4 and S5. Instead, non-covalent interactions between S4 and S5 have been suggested to be important for hyperpolarization activation of HCN channels[16,18–20]. A recent study suggested that S4 inhibits the PD of HCN channels when S4 is up and that the downward S4 movement at hyperpolarized voltages relieves this inhibition[16]. In another study, it was shown that a chimera with the VSD from a hyperpolarization-activated HCN channel and the PD from a depolarization-activated EAG channel could open both in response to depolarizations and hyperpolarizations[20]. The authors, therefore, suggested that S4 functions as a bipolar switch: S4 moves from a common resting position to an inward down position to open hyperpolarization-activated HCN channels, but S4 moves from the common resting position to an outward up position to open depolarization-activated EAG channels. However, our group has previously shown that S4 movement is similar in both depolarization-activated and hyperpolarization-activated ion channels, suggesting that the coupling between S4 and the gate is simply reversed in HCN channels compared with depolarization-activated Kv channels[3,10]. If this were the case, then one should be able to reverse the polarity of voltage gating in HCN channels by just modifying the interface between S4 and the pore.

Here we find that the most C-terminal region of S4 is required for both S4 movement and the coupling between S4 and the gate in HCN channels. We reverse the voltage dependence of HCN channels by mutating only two residues located at the interface between the C-terminal region of S4 and the PD such that the channels now open upon depolarization instead of hyperpolarization. Because the S4 movement in the mutant depolarization-activated HCN channels is similar to that in hyperpolarization-activated HCN channels, we suggest that these mutations reverse the coupling between S4 and the gate. Our data indicate that what determines whether HCN channels open by hyperpolarizations or depolarizations are small differences in the energies of the closed and open states, owing to different interactions between the voltage sensor and the pore in the different channels.

## Results

### The ΔQWE deletion inhibits S4 movement

It was recently shown that deleting the conserved sequence QWE (ΔQWE) in S4 of spHCN channels (Fig. 1a, b) in the presence of cAMP generates a voltage-independent open channel[16], as if S4 moves but is uncoupled from the gate or as if S4 is always in the down state

and the gate therefore is always open. To distinguish whether the ΔQWE channel is voltage independent because the movement of the S4 is uncoupled from the gate or because S4 does not move, we used voltage clamp fluorometry (VCF) to simultaneously measure S4 movement by the fluorescence from fluorophores attached to S4[9]. We first introduced the mutation R332C in S4 in the ΔQWE mutant and then labeled R332C/QWE (ΔQWE*) channels with Alexa-488-maleimide, (we indicate Alexa-488-labeled R332C channels with an *). We have previously shown that the fluorescence from Alexa-488-labeled R332C channels is a good reporter for S4 movement in the sea urchin HCN (spHCN) channels[9] (Fig. 1c). ΔQWE* channels display no voltage-dependent fluorescence changes (Fig. 1d, e), suggesting that the ΔQWE mutation inhibits S4 movement. It is possible that, in the ΔQWE* channel, S4 is in the down position even at depolarized voltages, which would open the normally closed gate providing that the coupling between S4 and the gate is intact. However, if in the ΔQWE* mutant channel S4 is immobilized in the up position, then the coupling between S4 and the gate would have to be disrupted to generate an open, voltage-independent channel. To test whether S4 is in the up or down position, we measured the accessibility of the R332C residue in the background of the ΔQWE mutant to extracellular tetramethyl-rhodamine C5-maleimide (TMRM) in solutions that keep the membrane at depolarized (in a high K+ solution) and hyperpolarized (in a low K+ "NMDG" solution) voltages. We have previously shown that R332C is accessible to membrane-impermeable cysteine reagents when S4 is up and that the accessibility is reduced when S4 moves down[3]. Indeed, the rate of TMRM labeling of R332C was higher in the high K+ solution than in NMDG solution, as if the accessibility of residue R332C depends on the state of S4 (Fig. 1f). In contrast, the rate of TMRM labeling of R332C/ΔQWE was similar in the depolarizing and hyperpolarizing solutions and similar to the labeling rate for R332C in the hyperpolarizing solution, as if S4 is in the down state in ΔQWE channels (Fig. 1f). However, it is unknown from these data whether the ΔQWE deletion also disrupts the coupling between S4 and the gate.

### QWE interactions stabilize the closed state with S4 up

To test whether the ΔQWE deletion disrupts the coupling between S4 and the gate, we used the cryo-EM structure of the human hHCN1 channel[7], which is a structure with a closed gate and S4 up, to create a homology model of the spHCN channel. In the spHCN model, QWE is located between S1 and S5 with W355 of the QWE motif pointing toward F216 in S1 and E356 in QWE pointing towards N370 in S5 (Fig. 2a). In addition, R367, one helical turn below N370 in S5, makes an electrostatic interaction with D471 in S6. The homologous residues to R367 and D471 in HCN2 have been suggested to stabilize the closed gate[21]. One possibility is that these interactions of QWE in S4 with S1 and S5 stabilize the closed gate when S4 is up by holding S5 in position to make the R367–D471 interaction to keep the channel closed. We mutated F216, W355, and N370, one at a time, to determine the roles of these residues in voltage gating of HCN channels. F216A* channels have a 12% conductance remaining at positive voltages, and W355N* channels and N370W* channels have 25% conductance remaining at depolarized voltages (Fig. 2b–e), suggesting that the F216A, W355N, and N370W mutations all destabilize the closed state relative to the open state when S4 is up. It has previously been shown that E356A channels have ~10% conductance remaining at depolarized voltages and that Q354A channels close normally at depolarized voltages[16]. This suggests that W355 contributes the most of the QWE motif to the stability of the closed state at depolarized voltages. One possibility why the E356A mutation does not destabilize as much

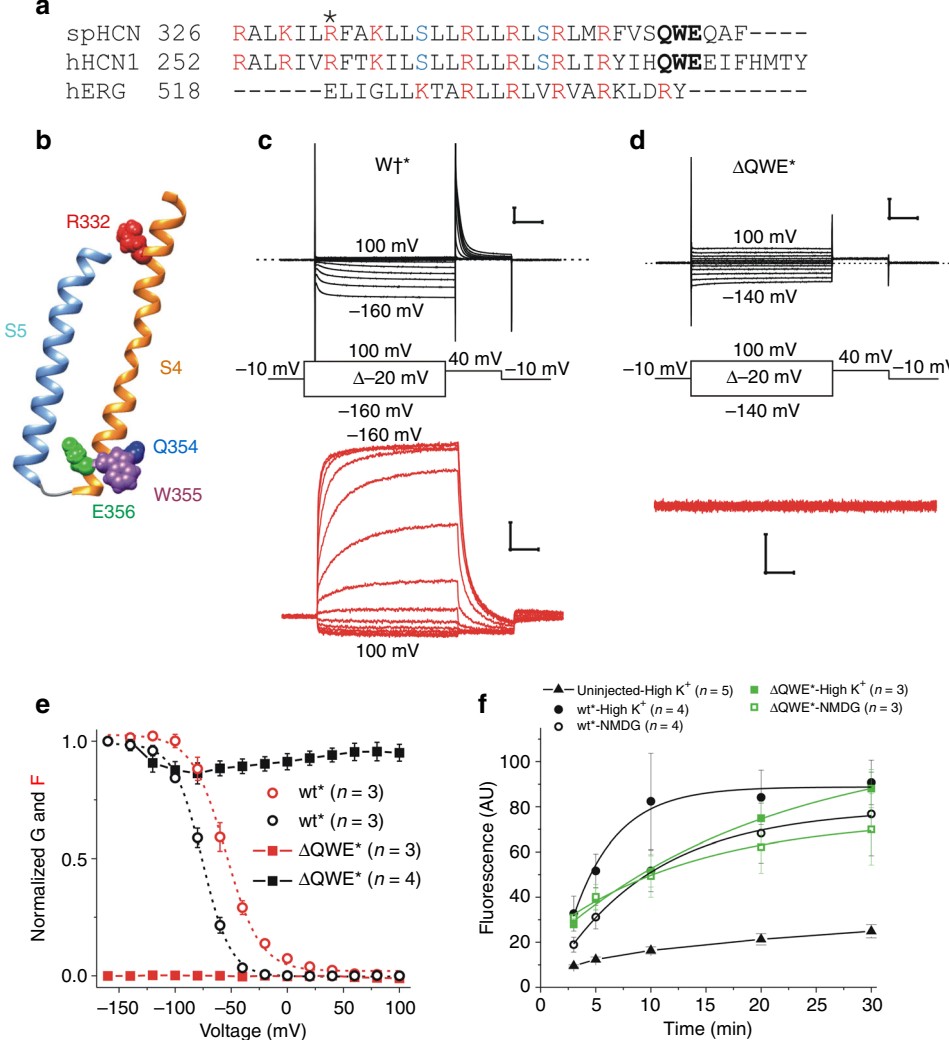

**Fig. 1 Deleting QWE at the C-terminal S4 leads to an always open channel with an immobilized S4. a** S4 sequence alignment of spHCN, hHCN1, and hERG channels. The conserved QWE sequence is shown in bold. R332 is shown with an *. **b** spHCN homology model showing the position of the QWE residues. **c**, Representative traces of the current (black) and fluorescence (red) from **c** wt* and **d** ΔQWE* channels. Scale bars: 0.1 s, 5 μA, and 2% ΔF/F. **e** Normalized $G(V)$ (black) and $F(V)$ (red) curves of wt* and ΔQWE* channels. **f** Tetramethyl-rhodamine C5-maleimide labeling of uninjected oocytes and oocytes expressing the R332C (wt) and R332C/ΔQWE (ΔQWE) channels in high $K^+$ and NMDG solutions. Labeling rates ($k$): 0.25 ± 0.06 (wt in high $K^+$), 0.11 ± 0.01 ($p = 0.05$) (wt in NMDG), 0.05 ± 0.01 ($p = 0.01$) (ΔQWE in high $K^+$), 0.07 ± 0.02 ($p = 0.001$) (ΔQWE in NMDG). Mean ± SEM. $p$ values indicate statistical difference from wt in high $K^+$ solution. Source data are provided as a Source Data file.

the closed state when S4 is up as N370W is that E356 might have additional interactions in the open state when S4 is up. Mutations of E356 would then not result in a net change in the relative stability of the closed and open states when S4 is up. Note that, at hyperpolarized voltages, the conductance decreases again for N370W* channels (Fig. 2c, e), suggesting that N370W also destabilizes the open gate when S4 is down. The destabilization of the open gate at hyperpolarized voltages for N370W is presumably owing to the removal of stabilizing interactions of N370 in the open state when S4 is down.

**Hydrophobic mutations at N370 disturb opening and closing**. To further study the role of N370, we mutated N370 to amino acids of different characteristics. Introduction of hydrophobic amino acids at position 370 generates leaky channels with ~50% conductance remaining at positive voltages and a decreasing conductance at extreme negative voltages (Fig. 3a–d, f). The fluorescence from most N370 mutants were similar to the fluorescence from wt* channels (but shifted around 30 mV for

N370A* and N370G*) (Fig. 3g), suggesting that the S4 undergoes a similar movement in these mutants as in wt* (just at slightly different voltages) and that instead the coupling between S4 and the pore is altered by the hydrophobic mutations at N370. In contrast, gate closing and opening was normal in N370E*, suggesting that a polar residue at 370 may be what is required for the normal gate opening and closing (Fig. 3e, f).

**W355N–N370W reverses the voltage dependence of activation**. Interestingly, the double mutant W355N–N370W* is a depolarization-activated channel (Fig. 4a–c). This shows that with only two single point mutations we can turn a hyperpolarization-activated HCN channel into a depolarization-activated channel. The kinetics and voltage dependence of the fluorescence signals from the depolarization-activated W355N–N370W* channels are very similar to those of hyperpolarization-activated HCN channels with a similar $V_{0.5}$ (Supplementary Fig. 1), as if the S4 movement is similar in these channels. The depolarization-activated W355N–N370W* channels are also blocked by the

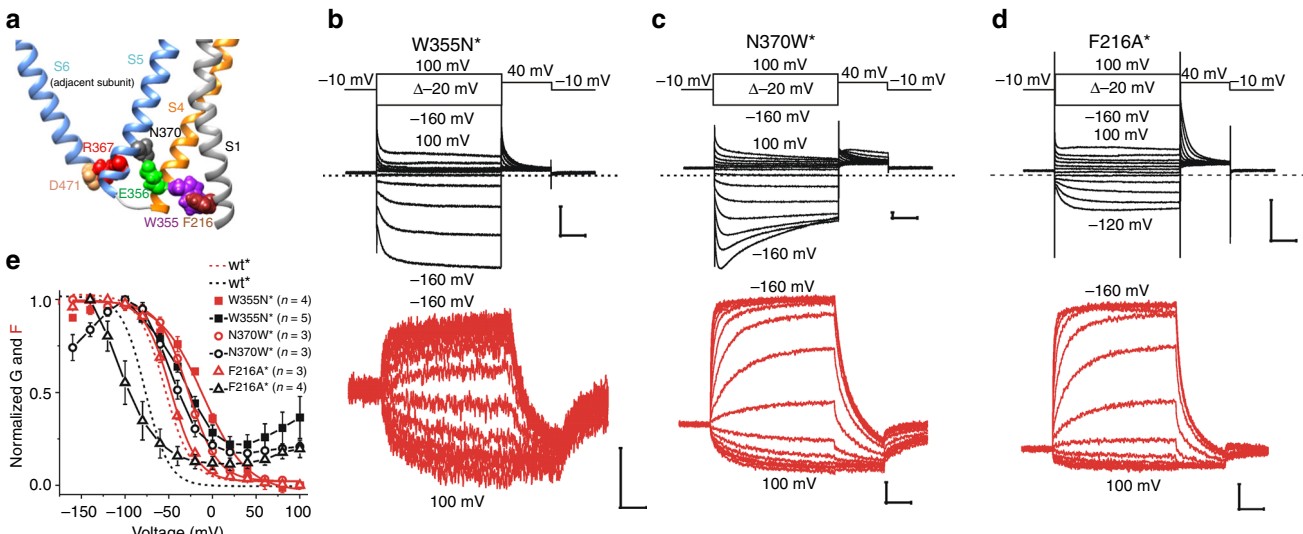

**Fig. 2 W355N*, N370W*, and F216A* single mutants do not close at positive voltages. a** spHCN homology model showing the position of W355, N370, and F216 with a closed gate and S4 up. **b–d** Representative current (black) and fluorescence (red) from **b** W355N*, **c** N370W*, and **d** F216A* channels. Scale bars: 0.1 s, 1 μA, and 1% ΔF/F. **e** Normalized G(V) and F(V) curves of W355N*, N370W*, and F216A* channels. G(V) and F(V) of wt* channels are shown for comparison (dashed lines). Mean ± SEM. Source data are provided as a Source Data file.

HCN channel pore blocker ZD7288 (Supplementary Fig. 2), suggesting that the mutations have not altered the pore. This all suggests that the W355N–N370W double mutation has turned a hyperpolarization-activated HCN channel into a depolarization-activated channel by simply altering the coupling between S4 and the pore, so that the open gate is destabilized relative the closed gate when S4 is down and the open gate is stabilized relative the closed gate when S4 is up. The opening of the W355N–N370W* channel at positive voltages is explained by the additive effects of the single mutants W355N* and N370W*, which both do not close completely at positive voltages. The closing of the W355N–N370W* channel at negative voltages is most likely an additive effect of each mutant to destabilize the open state with S4 down. For example, hydrophobic mutations of N370, such as N370W, destabilize the open state and cause a decrease in the currents at negative voltages (Fig. 3a–d). That hydrophobic mutations of N370 destabilize the open state is consistent with earlier data showing that in the open state the lower half of S4 is accessible to hydrophilic cysteine reagents, as if a water-filled crevice has opened up around lower S4[22]. We propose therefore, that in the open state with S4 down N370 faces lower S4 and would, therefore, also be in this water-filled crevice in the open state (Fig. 4d, left). Hydrophobic mutations of N370 would therefore destabilize the open state with S4 down.

## Discussion

It has previously been shown that HCN channels have a loose coupling[23], such that the gate can open either with S4 up or S4 down[24] (here shown as a simple four-state model of HCN channels in Supplementary Fig. 4). However, the open probability is much higher with S4 down than with S4 up, as if the free energies of the four different states (which depends on, e.g., the different residue–residue interactions in the different states) are such that the closed state is more stable than the open state when S4 is up and that the open state is more stable than the closed state when S4 is down. In general, proteins are held together in a specific conformation by a large number of stabilizing interactions, but relatively modest temperature increases denature most proteins[25]. This suggests that there is a delicate

balance of stabilizing and destabilizing forces to keep a protein in a specific conformation. Analogously, we show here that hyperpolarization-activated HCN channels can be turned into depolarization-activated channels by just two mutations, suggesting that there is a delicate balance of forces that makes HCN channels activated by hyperpolarization. Using existing structures and models of homologous channels[7,26,27], we created molecular models of spHCN channels in the different states to give a reasonable explanation for how the W355–N370W double mutation changes the energy balance between open and closed states in a way that switches hyperpolarization-activated HCN channels into depolarization-activated channels (Supplementary Figs. 3–5 and Supplementary Movies 1–2). In addition, these models give a simple mechanism for how wt* HCN channels are activated by hyperpolarization (Fig. 4d and Supplementary Fig. 4a). In these molecular models, there are several stabilizing interactions that keep the gate closed when S4 is up in wt HCN channels, including interactions formed by residues W355 and N370 (Fig. 4d, right and Supplementary Figs. 4a and 5). A downward movement of S4 in response to a hyperpolarization, as shown recently by FRET for spHCN[27] and by a S4 down cryo-EM structure of hHCN1[28], breaks these interactions and, instead, the gate opens when S4 moves down (Fig. 4d, left, and Supplementary Figs. 4a and 5 and Supplementary Movie 1). We speculate here that the opening up of the aqueous crevice around the hydrophilic N370 contributes to the stabilization of the open state with S4 down. The W355N–N370W* mutant removes the stabilizing interactions when the gate is closed and S4 is up allowing the W355N–N370W* mutant to open with S4 up (Fig. 4e, right and Supplementary Movie 2). The W355N–N370W* mutant also removes the stabilizing interactions (e.g., N370 with the aqueous crevice) when the gate is open and S4 down, making the W355N–N370W* mutant to close with S4 down (Fig. 4e, left and Supplementary Fig. 4b).

In conclusion, we here present a framework for voltage activation of HCN channels that will be essential to understand voltage activation in HCN and HCN-related channels, such as depolarization-activated hERG and EAG channels, and for further development of anti-arrhythmic and anti-epileptic drugs targeting HCN channels. For example, our model of the HCN

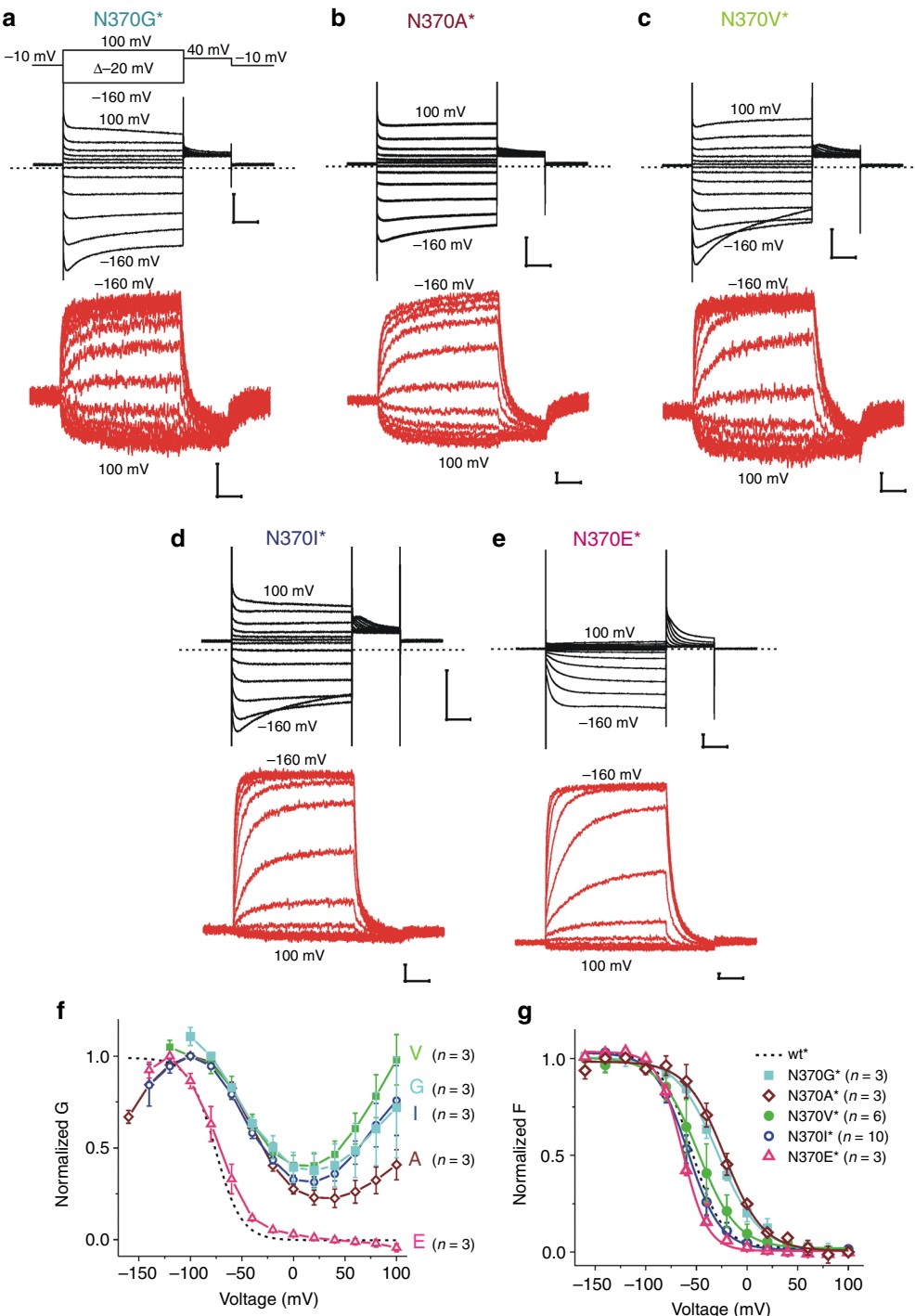

**Fig. 3 Hydrophobic N370 mutants have wt-like S4 movement but do not close completely.** Representative current (black) and fluorescence (red) from **a** N370G*, **b** N370A*, **c** N370V*, **d** N370I*, and **e** N370E* channels. Scale bars: 0.1 s, 5 µA, and 0.5% $\Delta F/F$. **f**, **g** Normalized **f** $G(V)$ and **g** $F(V)$ curves of wt*, N370G*, N370A*, N370V*, N370I*, N370D*, and N370E* channels. Mean ± SEM. Source data are provided as a Source Data file.

open state could be used to refine existing open-state blockers, such as ivabradine that is used to treat angina[29,30].

## Methods

**Molecular biology.** Point mutations were performed in the sea urchin (*Strongylocentrotus purpuratus*) HCN (spHCN) channel using site-directed mutagenesis (QuikChange Stratagene, CA, USA). The sequences of mutagenesis primers are listed in Supplementary Methods. We used the spHCN channel in this study because we have so far only been able to measure S4 movement with VCF in the spHCN channel. Deletions and replacement of amino acid segments were done using the In-Fusion technique (In-Fusion HD Cloning Kit, Takara Bio, CA, USA). The expression

plasmid for the spHCN channel was pGEM-HE. All the mutant DNAs were checked for correct sequences (Genewiz, NJ, USA). The DNA was linearized and cRNA was obtained using the T7 mMessage mMachine transcription kit (Ambion, TX, USA). RNA concentrations were measured using UV spectroscopy.

**Expression system.** The R332C mutation was introduced in all the mutant channels, unless otherwise noted, to be able to fluorescently label S4. For simplicity, we identified the mutant channels only by the mutation under study (omitting R332C) and an asterisk that represents S4 labeling at R332C. cRNA was injected at 1–4 µg/µl into the cytoplasm of stage IV and V *Xenopus laevis* oocytes (Ecocyte, Austin TX). The oocytes were incubated for 2–3 days at 18 °C for membrane expression.

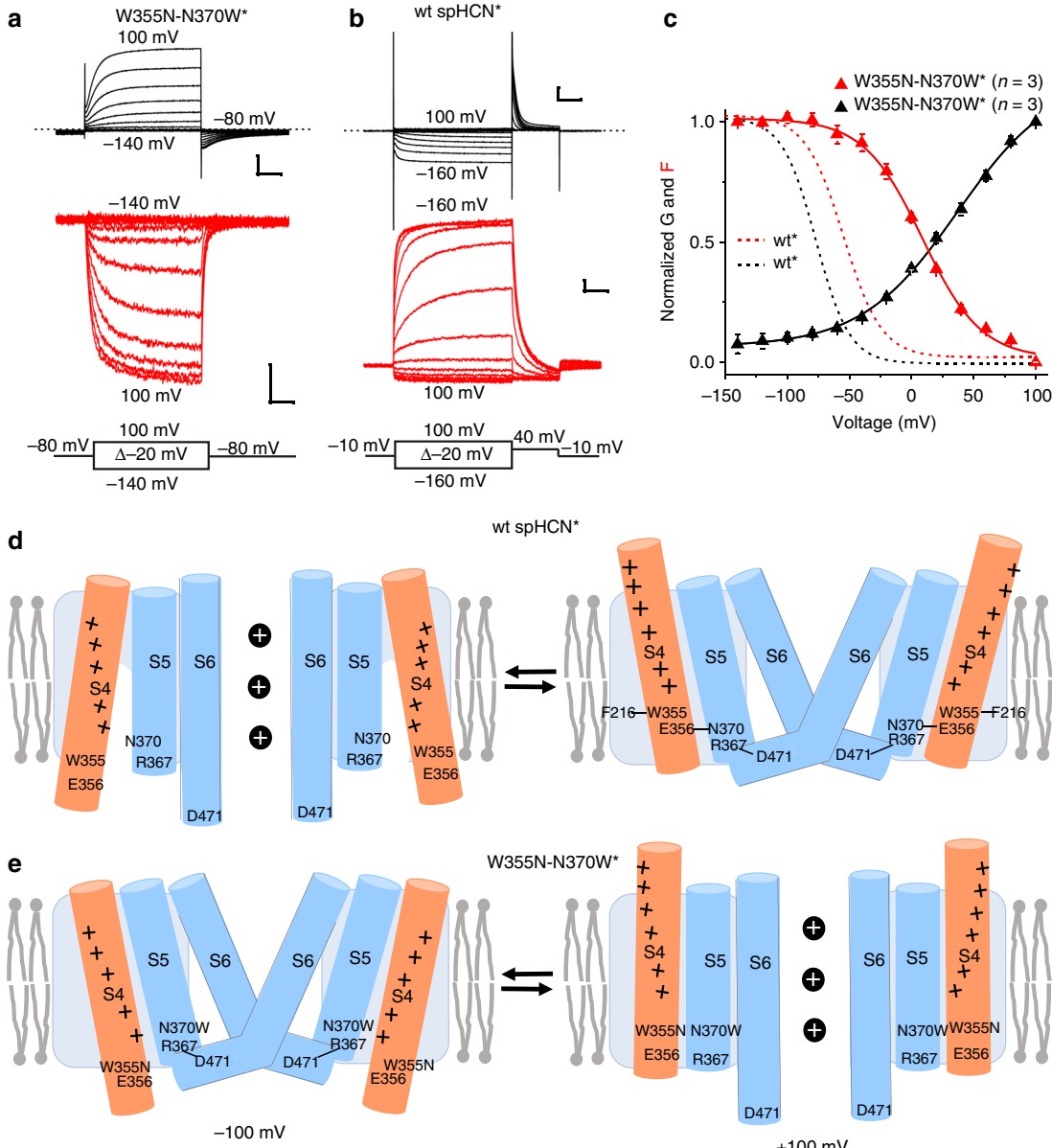

**Fig. 4 W355N–N370W switches the polarity of voltage activation. a, b** Currents (black) and fluorescence (red) from W355N–N370W* and wt spHCN* channels using holding voltages of **a** −80 mV and **b** −10 mV. Scale bars: 0.1 s, 5 μA, and 1% ΔF/F. **c** G(V) and F(V) curves of wt* and W355N–N370W* channels. Mean ± SEM. **d** W355-F216, N370-E356, and R367–D471 interactions hold wt* channels closed[7,21]. Downward S4 movement breaks these interactions and exposes N370 to water-filled crevice. **e** W355N–N370W removes stabilizing interactions in closed state with S4 up and destabilize water-filled crevice in open state with S4 down. See Supplementary Figs. 3–5 and Supplementary Movies 1–2 for more details. Source data are provided as a Source Data file.

**VCF**. The cells expressing the channels were labeled in a bath solution with 100 μM Alexa-488 C5-maleimide (Molecular Probes) for 30 minutes at 8 °C. After washing, oocytes were placed with the animal pole "up" in a bath housed on a Leica DMLFS upright fluorescence microscope. Light was focused on the animal pole of the oocyte through a ×20 objective (NA 1.0) and was passed through a filter cube from Chroma 41026 (HQ495/×30, Q515LP, HQ545/50 m). Note that, although the fluorescence is reported as a normalized value (ΔF/F), it is in most cases not possible to reliably compare the amplitude of the ΔF/F from mutation to mutation, owing to differences in the background oocyte fluorescence and channel expression. Whole-cell ionic currents were measured with the two-electrode voltage clamp technique using an Axon Geneclamp 500B amplifier (Axon Instruments, Inc.). Microelectrodes were pulled using borosilicate glass, filled with a 3 M KCl solution and had resistances between 0.5 and 0.9 MΩ. All experiments were performed at room temperature, but after labeling the oocytes were kept on ice to prevent internalization of labeled channels. Data were filtered at 1 kHz, digitized at 5 kHz (Axon Digidata 1322 A), and monitored and collected using pClamp software (Axon Instruments, Inc.). Fluorescence signals were low-pass Bessell filtered (Frequency Devices) at 200 Hz and digitized at

1 kHz. The noise of the fluorescence signal is highly dependent on the background fluorescence of the oocyte and the expression of the channels, which vary substantially from oocyte to oocyte. The bath solution (pH 7.5) contained in mM: 96 NaCl, 2 KCl, 1.8 MgCl2, and 5 HEPES. In addition, 100 μM LaCl3 was added to block oocytes endogenous currents. HCN currents were isolated by subtracting the currents in the presence of 1 mM of the HCN channel blocker ZD7288 (Tocris Bioscience, MN, USA).

**Cysteine accessibility**. The cells were labeled on ice with 10.5 μM TMRM in two different solutions. The high K+ solution contained (in mM): 100 KCl, 1.8 CaCl2, 5 NaCl, 10 HEPES, 1 MgCl2, and 2.5 Na–pyruvate. The low K+ "NMDG" solution contained (in mM): 96 NMDG, 2 mM KCl, 5 HEPES, 19 MgCl2, and 2.5 Na–pyruvate. The oocytes were washed and kept in ND96 after labeling. Fluorescence recordings of uninjected oocytes and oocytes expressing the different mutations were performed on the same day and same batch of oocytes with the same excitation light intensity. To calculate the rate of labeling (k) we fitted the points to a single exponential equation.

**Molecular modeling**. Homology models of closed and open spHCN channels with S4 up were created using the Swiss-model program with hHCN1 and hERG cryo-EM structures, respectively, as templates. Homology models of closed and open spHCN channels with S4 down were created by replacing the VSD in the spHCN models above with the recent VSD model of spHCN with S4 down[27]. The morph movies between the closed and open spHCN channel models were created in UCSF Chimera 1.13rc (San Francisco, CA, USA).

**Statistics and data analysis**. In all the figures, $n = 3$–10 cells. Average values are shown as mean ± SEM. The $G(V)$ and $F(V)$ curves were fitted to a Boltzmann equation using the Origin program (Origin 9.0.0, OriginLab Corporation). Statistical significance was tested with a one-way analysis of variance (Turkey's multiple comparisons test).

**Reporting summary**. Further information on research design is available in the Nature Research Reporting Summary linked to this article.

## Data availability

Data supporting the findings of this manuscript are available from the corresponding author upon reasonable request. A reporting summary for this Article is available as a Supplementary Information file. The source data underlying Figs. 1e–f, 2e, 3f–g, and 4c are provided as a Source Data file.

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

## Acknowledgements

This project was supported by NIH GM 109762 and HL 131461 to H. Peter Larsson. We thank Drs. R. Barro-Soria, K. Magleby, G. Dahl, S. Liin, S. Noskov, and J. Cui for comments and suggestions.

## Author contributions

R.R. and M.E.P. performed experiments. R.R. and H.P.L. designed experiments, analyzed and interpreted results, and wrote the manuscript.

## Competing interests

The authors declare no competing interests.
