## [Peer Review File · Nature Communications]

Reviewers' Comments:

Reviewer #1:

Remarks to the Author:

The study provides novel mechanistic insights into the gating of hyperpolarization-activated channels. The authors described in an earlier study that voltage sensor movement is conserved among hyper- and depolarization-activated channels and that the main difference between these type of channels is the voltage sensor to gate coupling. In the current study they report how this inverted coupling to the pore is achieved. The study is of major relevance for the understanding of HCN/SpIH, as well as hERG channel gating. Nevertheless some major questions remain:

1. Can the authors discuss why deltaQWE mechanistically works different than N370W or W355N and W355N/N370W? The deletion mutant has a destabilized open state with the voltage sensor fixed in the up state. N370W and W355N have a normal voltage sensor movement but a destabilized closed state and the W355N/N370W double mutant has a normal voltage sensor movement with an inverted coupling. I have problems understanding the three different phenotypes arising with mutations at more or less the same sites.
2. Why did the authors focus on N370W and W355N? Why not study mutations in the QWE motif at Q354 or E356, or the interacting residue F216 in S1? What happens when you mutate these residues or are mutations at this site not relevant?
3. Q354 seems to interact with D219 in Figure 2a. Why is this subsequently 'ignored' in the cartoons/model or why did the authors not test a mutation at D219?
4. A major point to me is, that the R367 to D471 interaction which is thought to stabilize the closed state of the channels is illustrated in Figure 2a, while this interaction does not seem to be present in the Supplemental Movies that describe the novel idea/model of HCN channel gating.
5. Page 4. The authors state: "There are several stabilizing interactions that keep the gate closed when S4 is up in wt HCN channels, including interactions formed by residues W355 and N370 (Fig. 4d, right and Suppl. Figs. S3a and S4)." The only interactions visualized are E356 to N370 and the previously described R367 to D471 salt bridge (missing in the movie). Does W355 to F216 and Q354 to D219 help stabilizing the closed state? As discussed above it is worth testing the respective mutants to elaborate the model, as currently the only new interaction site is E356 to N370 and W355 interacts with a non-defined "H".
6. Page 4. The authors state: "A downward movement of S4 in response to a hyperpolarization, as shown recently by FRET, breaks these interactions and, instead, other stabilizing interactions are formed when the gate opens and S4 is down (Fig. 4d, left, and Suppl. Figs. S3a and S4)." Which are the other interactions that are formed? Do you have data or something to speculate or include in the model?
7. The authors speculate at the end of the Abstract and the Results/Discussion section that the novel data and HCN channel gating model will help to develop antiarrhythmic and antiepileptic drugs. This is really vague and I have no idea how this could help. Can the authors be more precise, at least at the end of the manuscript or chose another main conclusion?

Reviewer #2:

Remarks to the Author:

The group of Peter Larson addresses an important question which is in the field since a while: Why the non-domain-swap channels HCN are activated by hyperpolarization whereas the Kv channels are activated by depolarization. This question have been extensively study without clear outcome.

In the present study Ramentol et al. use mutagenesis, TEVC and voltage-clamp fluorometry to address it. By mutating 2 residues, in the Ct part of the S4 domain which serves as an interactor with the pore domain, they have been able to reverse the polarity of HCN channels (i.e. from an hyperpolarized-channel to a depolarized-activated channel). This result demonstrates that the difference between channels which are activated by depolarization or hyperpolarization are just due some small differences in the free energy between closed and open states.

While these results are novel and important, there are some important issues that need clarification.

A. Ramentol and collaborators first compared the Kv10 and HCN channels since both channel families are non-domain-swapped channel families. QWE mutation in the Ct part of the S4 domain of HCN channels renders channel voltage independent. They found that the mutation suppressed the S4 movement explaining the absence of voltage dependency of the mutated channel. Next they looked if the S4 domain is locked in up or down state by looking at the cysteine accessibility (K335C) of this mutant. To do so, they labeled the oocyte injected with cRNA encoding for either wt or the mutated channel. They found that for both cases they found that oocytes were labeled by maleimide-Alex488.

1/ These experiments are done in high K condition that depolarizes the oocyte facilitating the up position for the WT. Could the authors show that the labeling of the WT is decreased in normal ND96 medium? And show that this does not modify the accessibility of the mutant as expected?

B. The authors then looked precisely the possible interacting residue of QWE motif and found that another important element should be the N370 which is localized R367 which is one turn below. To address the function of this interaction between S4 and S5 they mutated W355 and N370.

2/ Would it be possible to see the effect of the mutation of the E356 through a Ala and Asp to address also the importance of this charged residue?

C. Mutation of the N370 to Trp transform the channel into a leak channel without affecting the FV curve showing that S4 movement is not modified but the coupling yes.

D. when the N370 residue is replaced by hydrophobic residue, this increases the leaky part of the channel whereas a mutation through a Glu restores a normal opening and closing state of the channel.

3/ The VCF traces for the W355G mutant seem really noisy compared to the other, is it just an example or is it like that for all of the recordings (the current amplitude is not smaller)?

E. The double mutant W355N and N370W is now a depolarized-activated channel without affecting the S4 movement but by simply altering the coupling between VSD and PD demonstrating that the difference between channels which are activated by depolarization or hyperpolarization are just due some small differential interactions between VSD and PD.

F. The paper is clearly written but given that their no limitation space I would suggest to increase the conclusion part and add some information to help the reading such as:

Why did they choose the sea urchin clone?

Is this double mutant work the same for HCN2?

Reviewer #3:

Remarks to the Author:

General

This manuscript is based on a previous finding that deleting the conserved sequence QWE (354-

356) in the C-terminal part of the S4-helix of spHCN channels of the sea urchin generates a voltage-independent open channel (Flynn and Zagotta, reference 16). Using a homology model derived from HCN1 channels, the authors predicted for W355 and E356 the interaction partners F216 in the S1-helix and N370 in the S5-helix, respectively. Furthermore, a relevant interaction between R367 and D471 is predicted. Using a voltage-clamp fluorometry approach, the movement of the voltage sensor domain (VSD) was studied in parallel with the ionic currents.

Notably, single mutations W355N and N370W produced channels which showed not only voltage-activated currents at hyperpolarizing potentials but also at depolarizing potentials and, moreover, the double mutation reversed the channel from a hyperpolarization-activated to a depolarization activated channel. This result is indeed exciting.

By using cartoon models, the authors try to explain why structurally related HCN and Kv channels produce activation at hyperpolarization and depolarization, i.e. processes with opposite voltage dependence.

Though the topic of the manuscript is challenging, there are major limitations in both the data and the interpretation. Regarding the fluorescence measurements, the amplitude of $\Delta F/F$ varies considerably among the mutants (e.g. $\sim 1\%$ in W355N* and $\sim 6\%$ in N370W*, both at -160 mV). Because $\Delta F/F$ is already a normalized measure, this result should tell that the amount of S4 movement is significantly different among the different mutants. The authors did not consider this point at all. If including this aspect, the fluorescence traces are not anymore similar. Regarding the interpretation, it would be highly desirable to perform computations of free energies which go clearly beyond the level of cartoon models.

Specific

page 3, top sentence: The authors write: "It was recently shown that deleting the conserved sequence QWE (Δ QWE) in S4 (Fig. 1a-b) generates a voltage-independent open channel (reference 16)." In this reference, Flynn and Zagotta showed that in the absence of ligand, Δ QWE channels exhibited a voltage-dependent increase in conductance with depolarizing voltage steps. Only in the presence of cAMP there was an approximately voltage-independent gating.

End of second paragraph: It is unclear at which voltage the experiments were performed.

To confirm that the immobilization is in the up position, it would be helpful to show that labeling is indeed the same at depolarized and hyperpolarized voltages.

Second paragraph, Fig. 1f: It is not clear whether the expression is the same in K335C and K335C-QWE. If this is not clear the fluorescence intensity is not valid to say that there is a similar labeling in both mutant channels.

Third paragraph: There seems to be a correlation between the amplitude of the decay at hyperpolarization and the current amplitude at depolarization. The authors might like to follow this in more detail for their interpretations, in particular in the context of channel inactivation.

Third paragraph: The authors claim that the interactions W355-F216 and E356-R367 are relevant and they demonstrated effects by the mutations W355N and N370W, respectively. To confirm the specificity of the two interactions one expects also appropriate mutations in F216 and E356.

Fig. 2b,c: The currents look noticeably different, showing a pronounced decay (inactivation) for N370N* which is not present in W355N*. This difference deserves more attention in the context of the results.

page 3, bottom: The authors state: "The fluorescence from the different N370 mutants were very similar to the fluorescence from wt* channels (Fig. 3g)...". N370 wt is not similar to N370A,G,V as stated. There is a difference up to 50 mV. As already noted in the general remarks, the amplitude of fluorescence change $\Delta F/F$ is not similar in wt and mutant channels, e.g. 10% in wt (Fig. 1c) and 1.2% in N370G (Fig. 3a), which is unexplained. In a normalized signal a similar S4 movement should produce a similar fluorescence change.

It is not required to reproduce the traces from Fig. 1c in Fig. 3b.

page 4, line 9: Also here the examples of the fluorescence traces show that in W355N-N370W* the amplitude is only half of that in wt channels (see above), i.e. the amount of movement of the S4

helix is not similar.

page 4, end of second paragraph. The authors do not provide any evidence for the role of the water-filled crevice. This interpretation should therefore be declared as purely speculative.

page 4, begin of last paragraph: "It has previously been shown that HCN channels have a loose coupling²³, such that the gate can open either with S4 up or S4 down (here shown as a simple four-state model of HCN channels in Suppl. Fig. S3)."

The wording is misleading: In reference 23 it is not explicitly shown that the gate can open either with S4 up or down.

page 4, last paragraph: "However, the free energies of the four different states (which depends on, e.g., the different residue-residue interactions in the different states) are such that the open probability is much higher with S4 down than with S4 up. In general, proteins are held together in a specific conformation by a large number of stabilizing interactions, but relatively modest temperature increases denature most proteins²⁴." The sentence is not substantiated by free energies. How were they computed?

We thank the reviewers for their positive comments and constructive criticism. We have conducted new experiments and analysis, and we have revised the figures and text accordingly (shown in yellow highlight). We think the new data and the revised text have strengthened the manuscript and made it more clear. Below are detailed responses to the reviewers' comments.

Reviewer #1 (Remarks to the Author):

The study provides novel mechanistic insights into the gating of hyperpolarization-activated channels. The authors described in an earlier study that voltage sensor movement is conserved among hyper- and depolarization-activated channels and that the main difference between these type of channels is the voltage sensor to gate coupling. In the current study they report how this inverted coupling to the pore is achieved. The study is of major relevance for the understanding of HCN/SplH, as well as hERG channel gating. Nevertheless some major questions remain:

1. Can the authors discuss why deltaQWE mechanistically works different than N370W or W355N and W355N/N370W? The deletion mutant has a destabilized open state with the voltage sensor fixed in the up state. N370W and W355N have a normal voltage sensor movement but a destabilized closed state and the W355N/N370W double mutant has a normal voltage sensor movement with an inverted coupling. I have problems understanding the three different phenotypes arising with mutations at more or less the same sites.

- Note that the QWE deletion is always open and therefore has a destabilized closed state (not a destabilized open state, as the reviewer states), which is similar to N370W, W355N, and W355N/N370W that have a destabilized closed state when S4 is up. In addition, N370W has a tendency to close when S4 is down, which is further amplified in W355N/N370W. So those effects are similar in these two constructs containing the N370W mutation. S4 movement is similar in N370W, W355N, and W355N/N370W. So, the only major difference is that S4 is not moving in the QWE deletion. It is not surprising that this mutation is very different from the other mutations in the case of S4 movement, since in this mutation we remove three S4 residues.

2. Why did the authors focus on N370W and W355N? Why not study mutations in the QWE motif at Q354 or E356, or the interacting residue F216 in S1? What happens when you mutate these residues or are mutations at this site not relevant?

- Mutations E356A and Q354A were already reported in the earlier cited paper from Zagotta's lab (Flynn and Zagotta, 2017). Q354A closes normally but E356A has a 10% leak at positive voltages (which is less than the 25% leak of W355N and N370W). We now describe those previous data from Zagotta's lab. We now show data for F216A, which

also displays 12% leak (Fig. 2 d-e). Since W355 and F216 are interacting in our closed state model with S4 up, it makes sense that the mutations of these two residues have similar phenotypes. In our molecular model, E356 interacts with N370 in the closed state with S4 and R367 in the open state with S4 up. So mutating E356 would remove one stabilizing interaction in the closed state and one in the open state. This would explain why E356A does not produce as leaky channels as N370W, because E356A destabilizes both closed and open states equally. We have a second manuscript in preparation that describes the effects of E356A mutations and since it did not alter significantly the coupling, we would like to leave that mutation out of this paper and just refer to Zagotta's paper.

3. Q354 seems to interact with D219 in Figure 2a. Why is this subsequently 'ignored' in the cartoons/model or why did the authors not test a mutation at D219?

- Mutation Q354A did not introduced any leak currents (Flynn and Zagotta, 2017), so we have removed the Q354-D219 interaction in Fig 2a. We now think the important interactions are F216-W355 and E356-N370 in the closed state with S4 up.

4. *A major point to me is, that the R367 to D471 interaction which is thought to stabilize the closed state of the channels is illustrated in Figure 2a, while this interaction does not seem to be present in the Supplemental Movies that describe the novel idea/model of HCN channel gating.*

- Due to the fact that the R367-D471 interaction is an inter-subunit bond between neighboring subunits, it did not show up in the simple two-subunit movie with only two diametrically opposed subunits. We now show D471 and R367 from all four subunit to include this interaction in the movies.

5. *Page 4. The authors state: "There are several stabilizing interactions that keep the gate closed when S4 is up in wt HCN channels, including interactions formed by residues W355 and N370 (Fig. 4d, right and Suppl. Figs. S3a and S4)." The only interactions visualized are E356 to N370 and the previously described R367 to D471 salt bridge (missing in the movie). Does W355 to F216 and Q354 to D219 help stabilizing the closed state? As discussed above it is worth testing the respective mutants to elaborate the model, as currently the only new interaction site is E356 to N370 and W355 interacts with a non-defined "H".*

- See response to Comments 2 and 3 above. We now replaced H with F216 in the model figure.

6. Page 4. The authors state: "A downward movement of S4 in response to a hyperpolarization, as shown recently by FRET, breaks these interactions and, instead, other stabilizing interactions are formed when the gate opens and S4 is down (Fig. 4d, left, and Suppl. Figs. S3a and S4)." Which are the other interactions that are formed? Do you have data or something to speculate or include in the model?

- We agree that this was poorly worded. The open state with S4 down has not been solved for any channel, so we cannot make a very accurate homology model for this state. We previously just stated a hypothesis that there must be other bonds that stabilize the open state with S4 down. However, the fact that N370 hydrophobic mutations close again at hyperpolarized voltages supports that a hydrophilic environment for N370 is stabilizing the open state with S4 down. We now state that as a speculation for what stabilizes the open state with S4 down. We have now changed the text to: "We speculate here that the opening up of the aqueous crevice around the hydrophilic N370 contributes to the stabilization of the open state with S4 down".

7. The authors speculate at the end of the Abstract and the Results/Discussion section that the novel data and HCN channel gating model will help to develop antiarrhythmic and antiepileptic drugs. This is really vague and I have no idea how this could help. Can the authors be more precise, at least at the end of the manuscript or choose another main conclusion?

- We now have deleted that sentence from the Abstract. However, our model of the open state could be used as a tool to refine current open-state blocker (Ivabradine) that is used to treat angina. One could also use the open-state model to design small-molecule compounds that target the crevice that forms in the open state. These compounds could potentially trap the open state and lock the channel open to treat HCN channel mutations that have low open probability. We have added the first of these possibilities at the end of the Discussion.

Reviewer #2 (Remarks to the Author):

The group of Peter Larson addresses an important question which is in the field since a while: Why the non-domain-swap channels HCN are activated by hyperpolarization whereas the Kv channels are activated by depolarization. This question has been extensively studied without clear outcome. In the present study Ramentol et al. use mutagenesis, TEVC and voltage-clamp fluorometry to address it. By mutating 2 residues, in the Ct part of the S4 domain which serves as an interactor with the pore domain, they have been able to reverse the polarity of HCN

channels (i.e. from an hyperpolarized-channel to a depolarized-activated channel). This result demonstrates that the difference between channels which are activated by depolarization or hyperpolarization are just due some small differences in the free energy between closed and open states.

While these results are novel and important, there are some important issues that need clarification.

A. Ramentol and collaborators first compared the Kv10 and HCN channels since both channel families are non-domain-swapped channel families. QWE mutation in the Ct part of the S4 domain of HCN channels renders channel voltage independent. They found that the mutation suppressed the S4 movement explaining the absence of voltage dependency of the mutated channel. Next they looked if the S4 domain is locked in up or down state by looking at the cysteine accessibility (K335C) of this mutant. To do so, they labeled the oocyte injected with cRNA encoding for either wt or the mutated channel. They found that for both cases they found that oocytes were labeled by maleimide-Alex488.

1/ These experiments are done in high K condition that depolarizes the oocyte facilitating the up position for the WT. Could the authors show that the labeling of the WT is decreased in normal ND96 medium? And show that this does not modify the accessibility of the mutant as expected?

- K335C has a too negative voltage range of activation to reliably force all S4s into the down state using changes in the extracellular medium (high K⁺ versus low K⁺ solutions) during labeling. We therefore now use a different method to better measure the state-dependent accessibility, using the labeling rate of residue R332C with tetramethyl-rhodamine- C5 maleimide (TMRM) (Fig. 1f). R332C has a more positive voltage range of activation and is therefore easier to move its S4s into the down state using changes in the extracellular medium. We now show clear state-dependent accessibility of R332C for TMRM. The labeling rate for R332C-ΔQWE is state independent and similar to the rate for wt* in the S4 down state, suggesting that S4 in ΔQWE* channels are down.

B. The authors then looked precisely the possible interacting residue of QWE motif and found that another important element should be the N370 which is localized R367 which is one turn below. To address the function of this interaction between S4 and S5 they mutated W355 and N370.

2/ Would it be possible to see the effect of the mutation of the E356 through a Ala and Asp to address also the importance of this charged residue?

- E356A was earlier reported to have 10% leak currents (Flynn and Zagotta, 2017). See above answer to Reviewer 1's comment 2 for interpretation.

C. Mutation of the N370 to Trp transform the channel into a leak channel without affecting the FV curve showing that S4 movement is not modified but the coupling yes.

D. when the N370 residue is replaced by hydrophobic residue, this increase the leaky part of the channel whereas a mutation through a Glu restore a normal opening and closing state of the channel.

3/ The VCF traces for the W355G mutant seems really noisy compare to the other, is it just an example or it is like that for all of the recording (the current amplitude is not smaller)?

- We did not report on W355G. The reviewer is probably talking about N370G. The noise of the fluorescence signal is highly dependent on the background fluorescence of the oocyte and the expression of the channels, both which vary substantially from oocyte to oocyte. We now mention these caveats in the Methods section.

E. The double mutant W355N and N370W is now a depolarized-activated channel without affecting the S4 movement but by simply altering the coupling between VSD and PD demonstrating that the difference between channels which are activated by depolarization or hyperpolarization are just due some small differential interactions between VSD and PD.

*F. The paper is clearly written but view that their no limitation space I would suggest to increase the conclusion part and add some informations to help the reading such as:
Why did they choose the sea urchin clone?
Is this double mutant work the same for HCN2?*

- We used the spHCN clone, because this is the clone used in the original paper that reported the Δ QWE mutation (Flynn and Zagotta, 2017). It is also the best studied HCN clone and the only HCN clone for which S4 movement has been measured with voltage clamp fluorometry. We now discuss why we used the spHCN clone in the Methods.

We tested in mouse HCN2 the mutant homologous to spHCN-W355N-N370W. We found that this mutation did not produce a functional channel. Mammalian HCN channels are known to be more sensitive to mutations than spHCN channels, for which many mutations express but the homologous mammalian mutations do not express

(e.g. Vemana 2004, Bell 2004, Mannikko 2002).

Reviewer #3 (Remarks to the Author):

General

This manuscript is based on a previous finding that deleting the conserved sequence QWE (354-356) in the C-terminal part of the S4-helix of spHCN channels of the sea urchin generates a voltage-independent open channel (Flynn and Zagotta, reference 16). Using a homology model derived from HCN1 channels, the authors predicted for W355 and E356 the interaction partners F216 in the S1-helix and N370 in the S5-helix, respectively. Furthermore, a relevant interaction between R367 and D471 is predicted. Using a voltage-clamp fluorometry approach, the movement of the voltage sensor domain (VSD) was studied in parallel with the ionic currents.

Notably, single mutations W355N and N370W produced channels which showed not only voltage-activated currents at hyperpolarizing potentials but also at depolarizing potentials and, moreover, the double mutation reversed the channel from a hyperpolarization-activated to a depolarization activated channel. This result is indeed exciting.

By using cartoon models, the authors try to explain why structurally related HCN and Kv channels produce activation at hyperpolarization and depolarization, i.e. processes with opposite voltage dependence.

Though the topic of the manuscript is challenging, there are major limitations in both the data and the interpretation. Regarding the fluorescence measurements, the amplitude of $\Delta F/F$ varies considerably among the mutants (e.g. $\sim 1\%$ in W355N and $\sim 6\%$ in N370W*, both at -160 mV). Because $\Delta F/F$ is already a normalized measure, this result should tell that the amount of S4 movement is significantly different among the different mutants. The authors did not consider this point at all. If including this aspect, the fluorescence traces are not anymore similar. Regarding the interpretation, it would be highly desirable to perform computations of free energies which go clearly beyond the level of cartoon models.*

Specific

page 3, top sentence: The authors write: "It was recently shown that deleting the conserved sequence QWE (Δ QWE) in S4 (Fig. 1a-b) generates a voltage-independent open channel (reference 16)." In this reference, Flynn and Zagotta showed that in the absence of ligand, Δ QWE channels exhibited a voltage-dependent increase in conductance with depolarizing voltage steps. Only in the presence of cAMP there was an approximately voltage-independent gating.

- The reviewer is correct. We now state that in the presence of cAMP the QWE deletion made the spHCN channel almost completely voltage independent.

End of second paragraph: It is unclear at which voltage the experiments were performed. To confirm that the immobilization is in the up position, it would be helpful to show that labeling is indeed the same at depolarized and hyperpolarized voltages.

K335C has a too negative voltage range of activation to reliably force all S4s into the down state using changes in the extracellular medium (high K⁺ versus low K⁺ solutions) during labeling. We therefore now use a different method to better measure the state-dependent accessibility, using the labeling rate of R332C with tetramethyl-rhodamine-C5 maleimide (TMRM). R332C has a more positive voltage range of activation and is therefore easier to move its S4s into the down state using changes in the extracellular medium. We now show clear state-dependent changes in the labeling rate of R332C in wt* channels for TMRM. The labeling rate of ΔQWE* is state independent and similar to the labeling rate for wt* in the S4 down state, suggesting that S4 in the ΔQWE* channels are down.

Second paragraph, Fig. 1f: It is not clear whether the expression is the same in K335C and K335C-QWE. If this is not clear the fluorescence intensity is not valid to say that there is a similar labeling in both mutant channels.

We agree with the reviewer that the level of expression could affect the interpretation of this experiment. We thank the reviewer for pointing this out. We therefore changed from measuring the amount of labeling to measuring the rate of labeling, which is independent of the expression level. See response to previous comment.

Third paragraph: There seems to be a correlation between the amplitude of the decay at hyperpolarization and the current amplitude at depolarization. The authors might like to follow this in more detail for their interpretations, in particular in the context of channel inactivation.

- We are not sure we understand the comment, but we interpret the comments as concerning Fig. 3 and the fact that hydrophobic mutations that close at

hyperpolarized voltages also are leaky at depolarized voltages. We agree that this is a correlation in the sense that having a hydrophobic residue at N370 leads to these two phenomena. However, we think the underlying mechanisms are different in the two phenomena: The leaky current at depolarized voltages is due to the loss of a stabilizing interaction (i.e. the hydrogen bond between N370-E356) in the closed state with S4 up, whereas the closing at hyperpolarized voltages is due to the destabilizing effect of putting a hydrophobic residues (e.g. N370A) in the hydrophilic crevice in the open state with S4 down. These two phenomena occur in different conformations of the channels, one with S4 up and one with S4 down. So, we don't think these two effects are related more than they both rely on hydrophobic mutations at position N370.

Third paragraph: The authors claim that the interactions W355-F216 and E356-R367 are relevant and they demonstrated effects by the mutations W355N and N370W, respectively. To confirm the specificity of the two interactions one expects also appropriate mutations in F216 and E356.

- See answers above to Reviewer 1's comment 2.

Fig. 2b,c: The currents look noticeably different, showing a pronounced decay (inactivation) for N370N which is not present in W355N*. This difference deserves more attention in the context of the results.*

- We have expanded the discussion about this reclosing for N370W at hyperpolarized voltages in the manuscript.

page 3, bottom: The authors state: "The fluorescence from the different N370 mutants were very similar to the fluorescence from wt channels (Fig. 3g)....". N370 wt is not similar to N370A,G,V as stated. There is a difference up to 50 mV. As already noted in the general remarks, the amplitude of fluorescence change $\Delta F/F$ is not similar in wt and mutant channels, e.g. 10% in wt (Fig. 1c) and 1.2% in N370G (Fig. 3a), which is unexplained. In a normalized signal a similar S4 movement should produce a similar fluorescence change.*

- We agree that the previous sentence was not correctly stated. We have changed that sentence which now says : "The fluorescence from most N370 mutants were similar to the fluorescence from wt* channels (but shifted around 30 mV for N370A* and N370G*) (Fig. 3g), suggesting that the S4 undergoes a similar movement in these mutants as in wt* (just at slightly different voltages) and that instead the coupling between S4 and the pore is altered by the hydrophobic mutations at N370."

- The amplitude of the $\Delta F/F$ fluorescence signals depends on the expression of the channels (which contributes to both ΔF and F) and the endogenous fluorescence (which contributes only to F). Both the endogenous fluorescence (which is different in different oocytes due to color variations) and the expression of channels is highly variable from oocyte to oocyte. So it is not easy to correlate the amplitude of the $\Delta F/F$ to the amplitude of S4 displacement. Please, see the figure below that shows that even for the same channel (comparing two fluorescence recordings from wt channel that were performed on two different oocytes), fluorescence intensity varies from oocyte to oocyte.

- We only use the kinetics and voltage dependence when we compare the fluorescence of the different mutations. The voltage dependence is a more reliable measurement of the amplitude of S4 movement than the amplitude of the fluorescence. If the S4 movement really was only say 10% in N370G compared to wt then the voltage dependence of the fluorescence (the slope of the FV) would have to be only 10% of that in wt (because a 90%-reduced S4 amplitude would reduce the effective gating charge by 90%). But these channels have very similar voltage dependence and slope of the FV, as if S4 movement is very similar in these channels.

It is not required to reproduce the traces from Fig. 1c in Fig. 3b.

- The Reviewer probably means Fig 1c and 4b. We believe it helps the reader to see side-by-side comparison of wt and W355N-N370W.

*page 4, line 9: Also here the examples of the fluorescence traces show that in W355N-N370W**

the amplitude is only half of that in wt channels (see above), i.e. the amount of movement of the S4 helix is not similar.

See comment above about variation in fluorescence amplitude from oocyte to oocyte..

page4, end of second paragraph. The authors do not provide any evidence for the role of the water-filled crevice. This interpretation should therefore be declared as purely speculative.

- The idea of a crevice in the open state was the main conclusion of a prior study from Steve Siegelbaum's laboratory (Bell et al., JGP 2004), measuring cysteine accessibility with intracellular applied hydrophilic thiol reagents. The presence of an aqueous crevice around N370 fits with our data that hydrophobic mutations of N370 destabilize the open state and is consistent with our molecular model of the open state (Suppl. Figs. S4 and S5). We now have reworded this section and state: "We propose therefore, that in the open state with S4 down N370 faces lower S4 and would, therefore, also be in this water-filled crevice in the open state (Fig. 4d, left).".

page 4, begin of last paragraph: "It has previously been shown that HCN channels have a loose coupling²³, such that the gate can open either with S4 up or S4 down (here shown as a simple four-state model of HCN channels in Suppl. Fig. S3)."

The wording is misleading: In reference 23 it is not explicitly shown that the gate can open either with S4 up or down.

- The reviewer is correct that they did not state that explicitly, but from their model it is clear that the channel can open from either the C₀ state (all four voltage sensors are up) or the C₄ state (all four voltage sensors are down). However, we now add another reference that states explicitly that HCN channels can open with S4 down or up (Proenza et al., 2002). These authors showed that HCN2 channels have an open state at depolarized voltages and one open state at hyperpolarized voltages.

page 4, last paragraph: "However, the free energies of the four different states (which depends on, e.g., the different residue-residue interactions in the different states) are such that the open probability is much higher with S4 down than with S4 up. In general, proteins are held together in a specific conformation by a large number of stabilizing interactions, but relatively modest temperature increases denature most proteins²⁴." The sentence is not substantiated by free energies. How were they computed?

- Computations of free energies for spHCN channel models of the different states would be very difficult because we only have homology models of spHCN in the closed state with S4 up and the closed state with S4 down (based on the CryoEM structures of hHCN1). No structures of HCN channels are available of the other states (such as the open state and S4 down). Therefore, we did not compute free energies. The previous sentence was just a thermodynamic expression of the fact that HCN channels are biased to open (have a high open probability) when S4 is down and biased to close (have a small open probability) when S4 is up: i.e. the free energy of the open state when S4 is down must be less than the free energy of the closed state when S4 is down, because the channel spends more than 80% of its time open when S4 is down. Similarly, the free energy of the open state when S4 is up must be greater than the free energy of the closed state when S4 is up, because the channel spends more than 95% of its time closed when S4 is up. We now explain this better in the manuscript and have reworded the section as: "However, the open probability is much higher with S4 down than with S4 up, as if the free energies of the four different states (which depends on, e.g., the different residue-residue interactions in the different states) are such that the closed state is more stable than the open state when S4 is up and that the open state is more stable than the closed state when S4 is down."

Reviewers' Comments:

Reviewer #1:

Remarks to the Author:

Thank you for the great revision and the additional information/data/explanations. As the authors have addressed all my concerns, I have no further or additional criticisms. Great work.

Reviewer #2:

Remarks to the Author:

1. The authors have done a great job in addressing the concerns raised in the first round of review. I support the publication of this work in 'nature Communication'. I think it has profound implications for channel function and will have a major impact on the field.

2. There are still a few minor issues with the manuscript that should be addressed:

The authors improved the methods, but some key facts are still missing. The amounts of RNA used to inject oocytes are missing. Please include (ng, μ g, mg, kg? How much of the nucleic acid polymer was used). Such details are essential for addressing reproducibility of both the electrophysiology and VCF experiments.

Reviewer #3:

Remarks to the Author:

The revised version of the manuscript has significantly improved and adequately addressed major criticisms of the other two reviewers and my own. The impact of the results will probably stimulate the work in the field.

From my point of view two points deserve further consideration:

1. Regarding TMRM labeling of R332C the authors now use the presumably expression-independent rate of labeling instead of measuring the fluorescence intensity. They claim by the new Fig. 1f that for R332C the rate of TMRM labeling in K⁺ solution was higher than that in NMDG solution whereas the rates in R332C/ Δ QWE were similar and similar to that for R332C in NMDG solution. If so this argument would support the story. However, the difference in the rates for R332C with the two solutions requires a statistical test because the conclusion seems to be substantiated by a single data point at 10 min only and the error bars (s.e.m.) are very large.
2. The authors emphasize a noticeable variability among the oocytes concerning the amplitude of the $\Delta F/F$ signals which depend on the expression of the channels (which contributes to both ΔF and F) and the endogenous fluorescence (which contributes only to F). To underline this, the authors present an example of two oocytes showing that even for the same wt channels $\Delta F/F$ varies by more than tenfold. If the variability of this already relative measure is so strong, the authors should discuss this in more detail with regard to their results.

Response to REVIEWERS' COMMENTS

Reviewer #1 (Remarks to the Author):

Thank you for the great revision and the additional information/data/explanations. As the authors have addressed all my concerns, I have no further or additional criticism. Great work.

Reviewer #2 (Remarks to the Author):

1. The authors have done a great job in addressing the concerns raised in the first round of review. I support the publication of this work in 'nature Communication'. I think it has profound implications for channel function and will have a major impact on the field.

2. There are still a few minor issues with the manuscript that should be addressed:

The authors improved the methods, but some key facts are still missing. The amounts of RNA used to inject oocytes are missing. Please include (ng, μ g, mg, kg? How much of the nucleic acid polymer was used). Such details are essential for addressing reproducibility of both the electrophysiology and VCF experiments.

- *We now include the concentration of RNA injected in Methods.*

Reviewer #3 (Remarks to the Author):

The revised version of the manuscript has significantly improved and adequately addressed major criticisms of the other two reviewers and my own. The impact of the results will probably stimulate the work in the field.

From my point of view two points deserve further consideration:

1. Regarding TMRM labeling of R332C the authors now use the presumably expression-independent rate of labeling instead of measuring the fluorescence intensity. They claim by the new Fig. 1f that for R332C the rate of TMRM labeling in K⁺ solution was higher than that in NMDG solution whereas the rates in R332C/ Δ QWE were similar and similar to that for R332C in NMDG solution. If so this argument would support the story. However, the difference in the rates for R332C with the two solutions requires a statistical test because the

conclusion seems to be substantiated by a single data point at 10 min only and the error bars (s.e.m.) are very large.

- *We used an ANOVA statistical test for the rate constants. The rate constants for the mutant were found to be significantly different from the rate constant for wt in depolarizing solutions. This is listed in the Figure 1 legend.*

2. The authors emphasize a noticeable variability among the oocytes concerning the amplitude of the $\Delta F/F$ signals which depend on the expression of the channels (which contributes to both ΔF and F) and the endogenous fluorescence (which contributes only to F). To underline this, the authors present an example of two oocytes showing that even for the same wt channels $\Delta F/F$ varies by more than tenfold. If the variability of this already relative measure is so strong, the authors should discuss this in more detail with regard to their results.

- *We mention this in the Methods sections “Note that, although the fluorescence is reported as a normalized value ($\Delta F/F$), it is in most cases not possible to reliably compare the amplitude of the $\Delta F/F$ from mutation to mutation, due to differences in the background oocyte fluorescence and channel expression.” and “The noise of the fluorescence signal is highly dependent on the background fluorescence of the oocyte and the expression of the channels, which vary substantially from oocyte to oocyte.”. Neither of these variable factors, fluorescence amplitude and fluorescence noise, impact our conclusions reached in this manuscript.*